# The Human Genetic Differences in the Outcomes of mRNA Vaccination against COVID-19: A Prospective Cohort Study

**DOI:** 10.3390/vaccines12060626

**Published:** 2024-06-05

**Authors:** Ha-Eun Ryu, Jihyun Yoon, Ja-Eun Choi, Seok-Jae Heo, Kyung-Won Hong, Dong-Hyuk Jung

**Affiliations:** 1Department of Family Medicine, Yonsei University College of Medicine, Seoul 03722, Republic of Korea; rahahaha@yuhs.ac; 2Department of Family Medicine, Anam Hospital, Korea University College of Medicine, Seoul 02481, Republic of Korea; 3R&D Division, Theragen Health Co., Ltd., Pangyoyeok-ro, Bundang-gu, Seongnam-si 13493, Republic of Korea; 4Division of Biostatistics, Department of Biomedical Systems Informatics, Yonsei University College of Medicine, Seoul 03722, Republic of Korea

**Keywords:** GWAS, HDAC9, COVID-19, mRNA vaccine, humoral immunity

## Abstract

Background: This study aimed to explore how genetic variations in individuals impact neutralization activity post-mRNA vaccination, recognizing the critical role vaccination plays in curbing COVID-19 spread and the necessity of ensuring vaccine efficacy amidst genetic diversity. Methods: In a 4-week clinical pilot study, 534 healthy subjects received their first COVID vaccine dose, followed by the second dose. Antibody levels were evaluated thrice. From this pool, 120 participants were selected and divided into high- and low-antibody groups based on their levels. Genomic DNA was isolated from peripheral blood mononuclear cells for pilot genome-wide association studies (GWAS) conducted on a single platform. Real-time PCR was used to confirm differences in gene expression identified via GWAS analysis. Results: Three SNPs exceeded the level of *p* < 1.0 × 10^−3^. The rs7795433 SNP of the HDAC9 gene (7q21.1) showed the strongest association with COVID-19 vaccination under the additive model (OR = 5.63; *p* = 3 × 10^−5^). In the PCR experiments, the AA genotype group showed that the gene expression level of HDAC9 was likely to be decreased in the low-antibody-formation group at the time of vaccination. Conclusion: We found that AA genotype holders (rs7795433 SNP of the HDAC9 gene) have a high probability of having a higher antibody count when vaccinated, and GG type holders have a high probability of the opposite. These findings show that the genetic characteristics of vaccinated people may affect antibody production after COVID vaccination.

## 1. Introduction

The global impact of coronavirus disease 2019 (COVID-19), resulting from severe acute respiratory syndrome coronavirus 2 (SARS-CoV-2), has triggered widespread illness and mortality, raising concerns about global susceptibility to infectious diseases [1]. To address this pandemic, there has been vigorous research into vaccines, and the introduction of mRNA-based vaccines appears to be quite effective in managing it; numerous clinical studies have shown their effectiveness in providing protection and ensuring safety [2,3,4,5].

The rapid global approval and deployment of vaccines in response to the global pandemic marked an unprecedented effort in the fight against COVID-19 [6,7]. Despite the success of widespread vaccine administration, variations in immune responses have been observed among individuals [8,9]. Factors such as environmental influences, pre-existing health conditions, and social circumstances, along with genetic factors, have been proposed as host-related contributors to this variability [10,11,12]. Within this multifaceted landscape, our particular focus is on investigating the role of host genetic elements. This holds significance in developing a deeper understanding of the mechanisms underlying COVID-19 and in more clearly distinguishing modifiable factors.

Genetic polymorphisms are recognized to impact responses to both viral infections and immunization. In fact, previous studies have revealed that with respect to conventional vaccines, such as MMR, HBV, and influenza, the reasons for individual variations in immune responses after vaccination can be significantly explained by differences in genetic factors [13,14,15,16]. The influence of host genetic factors on COVID-19 infection and vaccination is also garnering attention [17]. The human leukocyte antigen (HLA) gene, a key player in the immune response, has the potential to influence T-cell immune responses related to antigen presentation and the establishment of lasting immune memory [18]. Genetic polymorphisms in HLA, based on genome-wide association studies (GWAS), have been reported to impact the clinical progression of patients infected with RNA viruses, such as SARS-CoV-2. Notably, HLA-DRB1*04:01, HLA-B*46:01, and HLA-C*04:01 have been associated with severe clinical outcomes or protective effects in COVID-19 [19,20,21].

In spite of the active research and understanding surrounding COVID-19, our comprehension of host genetic factors influencing the generation and maintenance of antibodies remains limited. Recognizing the importance of understanding the link between individual immune genetic factors and host immune responses, we aimed to explore the genetic factors within the Korean population that influence differences in neutralizing antibody production after mRNA vaccination against COVID-19.

## 2. Materials and Methods

### 2.1. Study Design and Participants

This was a longitudinal study of the genetic effects on COVID-19 antibodies after vaccination in Korean adults (Clinical Research Information Service, KCT0007342). We recruited a total of 534 subjects who received their first COVID-19 vaccine in August 2021. All subjects completed three visits. Written informed consent was obtained from all patients prior to participation in accordance with the Declaration of Helsinki, and with approval from the Institutional Review Board of Yongsebrans Hospital (IRB No. 9-2021-0101). All subjects who received the second dose were followed up at 2 weeks and 4 weeks to measure antibody responses. While there was no significant difference in antibody levels between participants after the first vaccination, a notable discrepancy in antibody levels was observed between subjects two weeks after the second vaccination. Based on this antibody response, we created two groups: a high-antibody group and a low-antibody group. Each group consisted of 60 subjects: the high-antibody group were the 60 patients who showed the highest percentage increase in antibody titres two weeks after the second dose compared to their antibody titres after the first dose; and the low-antibody group were the 60 patients with the lowest percentage increase in antibody titers 2 weeks after the second dose compared to their antibody titers after the first dose. Among these 120 selected subjects, 14 subjects who received a viral-vector-based COVID-19 vaccine instead of an mRNA COVID-19 vaccine at dose 2 were excluded, resulting in 50 subjects in the high-antibody group and 56 subjects in the resistant group (Figure 1).

### 2.2. Measurement of Anthropometric and Biochemical Parameters

Participants were examined a total of three times: at baseline, i.e., after completion of the first vaccine dose only; two weeks after the second COVID-19 vaccine dose; and four weeks later. At each visit, weight, height, systolic blood pressure (SBP), and diastolic blood pressure (DBP) were measured, and blood samples were taken. Participants were weighed and measured for height in light clothing and without footwear. Heights were recorded to the nearest 0.1 cm using a Seca 225 (SECA, Hamburg, Germany), and weights were determined to the nearest 0.1 kg on a GL-6000-20 scale (G-tech, Seoul, Republic of Korea). Body mass index (BMI) was subsequently calculated as the weight in kilograms divided by the square of height in meters (m^2^). SBP and DBP were recorded in the seated position using a Heine Gamma^®^ G7 aneroid sphygmomanometer (Heine Optotechnik, Hessing, Germany) following at least a five-minute rest. Blood samples were taken after a minimum of 8 h of fasting and were analyzed for plasma glucose, total cholesterol, triglycerides, high-density lipoprotein (HDL) cholesterol, low-density lipoprotein (LDL) cholesterol, alanine aminotransferase (ALT), and aspartate aminotransferase (AST) using a Cobas 8000 c702 module (Roche Diagnostics, Mannheim, Germany). White blood cell (WBC) count was measured with an XN-9000 (Sysmex Corporation, Kobe, Japan). Levels of 25-hydroxyvitamin D (vitamin D) were assessed using the Cobas 8000 e801 module (Roche Diagnostics). Total immunoglobulin E (IgE, reference range, ≤100 kU/L) was quantified using the Phadia 250 (Phadia, Uppsala, Sweden). Hypertension was defined as SBP ≥ 140 mm Hg, DBP ≥ 90 mm Hg, or current use of antihypertensive medication, and type 2 diabetes was defined as previous diagnosis of type 2 diabetes or fasting glucose ≥126 mg/dL.

### 2.3. Detection of Virus-Specific Antibodies

Automated ECLIA tests were performed using two types of SARS-CoV-2 antibody kits on the Cobas 8000 e801 module (Roche Diagnostics, Mannheim, Germany). The Elecsys Anti-SARS-CoV-2 assay uses a recombinant protein representing the nucleocapsid (N) antigen for the qualitative detection of antibodies to SARS-CoV-2, with the results interpreted as negative for anti-SARS-CoV-2 antibodies if the cut-off index (COI) is less than 1.0 and as positive if it is 1.0 or higher. The Elecsys Anti-SARS-CoV-2 S assay uses a recombinant protein representing the receptor-binding domain (RBD) of the spike (S) protein for the quantitative determination of antibodies to SARS-CoV-2, with results below 0.80 U/mL considered negative and results of 0.80 U/mL or above considered positive. Surrogate virus neutralization tests (sVNT) were performed using the cPass SARS-CoV-2 neutralization antibody detection kit (GenScript, Piscataway, NJ, USA) in conjunction with the SpectraMax 190 microplate reader (Molecular Devices, San Jose, CA, USA). This reader, optimized for DNA analysis in a 96-well format, also excels in the accurate quantification of enzyme-linked immunosorbent assays (ELISA). It uses a photometric system capable of detecting low concentrations of biological samples, ensuring accurate measurement of neutralizing antibodies against SARS-CoV-2. Results were interpreted as percentage inhibition (%inhibition) based on OD_450_ intensity. The manufacturer-recommended cut-off of ≥30% signal reduction was used to indicate the presence of anti-SARS-CoV-2 neutralizing antibodies. All %inhibition results were converted to IU/mL of the WHO International Standard using an Excel-based conversion tool [22]. The upper limit of the measurable range was 97.57% inhibition (or 3002 IU/mL). All tests were processed according to the manufacturer’s instructions.

### 2.4. Genome-Wide Association Study

The genomic DNA used in this study was isolated from peripheral blood mononuclear cells. A pilot GWAS was performed using a Theragen Precision Medicine Research Array (PMRA array), a customized array based on the Asian Precision Medicine Research Array, to genotype study subjects (cases) and controls on a single platform (Thermo Fisher Scientific, Waltham, MA, USA). From the resulting data, markers with Hardy–Weinberg equilibrium *p*-values less than 10^−6^, totaling 20,071; genotype call rates less than 97%, totaling 251,155; and minor allele frequencies (MAF) of 0.01, totaling 210,965, were sequentially discarded. This left 301,925 SNPs available for subsequent analysis. We then performed principal component analysis (PCA) to identify the first principal component (PC1) and the second principal component (PC2), which explained most of the variation in the data (see Appendix A).

### 2.5. Real-Time PCR

cDNA was synthesized using the ReverTra Ace qPCR RT Kit (Toyobo, Osaka, Japan) according to the manufacturer’s recommendations. PCR reactions were performed on a QuantStudio 12K Flex (https://www.thermofisher.com/order/catalog/product/4472048) Real-Time PCR System (ThermoFisher, Waltham, MA, USA) in 384-well plates with a total volume of 10 µL per reaction. The reaction mixture included Universal Master Mix, dNTPs, MgCl_2_, and AmpliTaq Gold from Applied Biosystems with 0.5 µL of a 20× TaqMan Assay for the HDAC9 gene (assay ID: Hs01081558_m1; Applied Biosystems, Waltham, MA, USA). GAPDH was used as an internal control using the TaqMan Human GAPDH Assay (assay ID: Hs99999905_m1; Applied Biosystems, Waltham, MA, USA). Each reaction mixture also contained 2 µL of template cDNA and 2.5 µL of distilled water. Amplification conditions were set to an initial denaturation at 95 °C for 10 min, followed by 40 cycles of 95 °C for 15 s and 60 °C for 1 min. Each sample was amplified in triplicate, and data analysis was performed using QuantStudio 12K Flex software.

### 2.6. Statistical Analysis

The association between case–control status and individual SNPs was assessed using odds ratios (ORs) and *p*-values. Multivariate adjustment was performed using two models: Model 1 included age, gender, and body mass index (BMI) as covariates; while Model 2 extended Model 1 by including vaccine type and the first and second principal components (PC1 and PC2) derived from principal component analysis. We conducted logistic regression analyses using PLINK (ver. 1.9, NIH-NIDDK’s Laboratory of Biological Modeling, the Purcell Lab, and others). Locus plots were generated for genome-wide significant loci using LocusZoom (ver. v0.4.8, University of Michigan, Department of Biostatistics, Center for Statistical Genetics [23] ).

### 2.7. Power Calculation

Adjusted for an assumed α of 0.05 and a power of 0.8, we performed power and sample size calculations for genetic association studies while considering the potential impact of mis-specifying the genetic model. We employed the genpwr package within the R statistical program for this analysis [24,25,26]. We defined the true genetic model as an additive model, reflecting the genuine relationship between genotype and the outcome variable. Additionally, we specified a ‘Test’ model to indicate how the genetic effect would be encoded for the purpose of statistical testing. Consequently, the number per group was determined to be 32 in Model 1 and 25 in Model 2 (refer to Appendix A). Finally, the number of subjects in each group was determined to be 60 in consideration of various conditions, such as papers on similar research topics, the statistical subject calculation process, and securing blood samples from the subjects [27].

## 3. Results

### 3.1. Population Characteristics

For this study, a total of 534 participants who received two doses of the COVID-19 vaccine were recruited and stratified into either a high-antibody group (n = 60) or a low-antibody group (n = 60) in a 1:1 ratio based on their neutralizing antibody titers (anti-N-Ab). To eliminate the effects of vaccine type differences, individuals who received a viral vector vaccine at either dose (n = 14) were excluded from the analysis. Table 1 shows the baseline characteristics of the subjects in each group. The high- and low-antibody groups had comparable characteristics with respect to age, the presence of diabetes, and rates of hypertension. Measures of fasting plasma glucose, total cholesterol, liver enzymes, vitamin D, and IgE were not significantly different between the groups. However, participants in the low-antibody group had lower rates of obesity and lower white blood cell (WBC) counts than those in the high-antibody group. Figure 2 illustrates the typical pattern of antibody response: anti-N-Ab was generated after the first vaccine dose, with a significant increase observed for two weeks after the second dose. Antibody levels declined from their peak until four weeks after the second dose.

### 3.2. GWAS Results

We used logistic regression analysis to identify statistically significant associations with COVID-19 vaccination. Unfortunately, none of the GWAS results achieved conventional genome-wide significance (*p* < 5 × 10^−8^). Therefore, we applied the study-wise criteria (*p* < 0.001), and 80 SNPs passed this study criteria with adjustment for three factors (age, gender, body mass index). As shown as Table 2, the rs7795433 SNP of the *HDAC9* gene (7q21.1) showed the strongest association with COVID-19 vaccination under additive model (OR = 5.63; *p* = 3 × 10^−5^). Genotype and allele frequencies of rs7795433 were analyzed within the antibody groups (1 = high-antibody; 0 = low-antibody). As a result of the analysis, it was found that the high-antibody group had a large distribution of AA genotype or A allele; and on the contrary, the low-antibody group had a large distribution of the GG genotype or G allele.

### 3.3. Sample Distribution and Allele Frequency by Genotype of HDAC9 SNP (rs7795433)

As shown in Table 3, we analyzed the genotype and allele frequencies within the antibody groups (1 = high-antibody; 0 = low-antibody). The high-antibody group had a lot of AA genotypes or A alleles; whereas the low-antibody group had a lot of GG genotypes or G alleles. Through this result, we found that AA genotype holders have a high probability of having an antibody higher when vaccinated, and GG type holders have a high probability of the opposite.

### 3.4. Real-Time PCR Analysis of HDAC9 SNP (rs7795433)

Table 4 describes the average delta ct (PCR ct value corresponding to the relative expression level, comparing the expression levels of the candidate gene Hdac9 and the house keeping gene GAPDH gene) for each sample. The higher the average delta value, the higher the ct value, which means that the PCR pick appears late, meaning that the amount of HDAC9 in the initial sample is low. The AA genotype group showed that the gene expression level of HDAC9 was likely to be decreased in the low-antibody-formation group at the time of vaccination. However, in this result, it is judged that a large amount of RNAs were decomposed because the samples for the RNA seq were tested in a frozen state. As a result, the average ct value of blood samples for GAPDH expression were 16–17; whereas in this study, a high ct value of above 35 was observed. This is an important limitation of this study, meaning that the real-time PCR results should only be used as reference values and not as the main results of this study.

## 4. Discussion

In this study, we divided individuals into high- and low-antibody titers after SARS-CoV-2 vaccination and conducted a GWAS to analyze genetic differences. The results revealed the most significant difference in the HDAC9 gene’s SNP rs7795433, identifying it as a promising candidate gene influencing antibody production differences. Additionally, when examining the allele frequency of the HDAC9 SNP, the A allele and A allele carriers (AA + GA) were more likely to be in the high-antibody group, while the G allele and G allele carriers (GA + GG) were more likely to be in the low-antibody group. Furthermore, RT-PCR was performed on three individuals from each AA, GA, GG genotype, and antibody group to confirm HDAC9 gene expression differences. The results show that in the high-antibody group, the AA genotype had the highest expression, while in the low-antibody group, the GG genotype had the highest expression. This suggests an association between better antibody production and the AA genotype. Additionally, when comparing antibody groups within the same genotype, significant differences in HDAC9 gene expression were observed, with AA (*p* = 0.011) and GA (*p* = 0.049) genotypes showing significantly higher expression in the high0antibody group compared to the low-antibody group. This implies that higher HDAC9 activity may contribute to increased antibody production.

Histone deacetylases (HDACs) are enzymes crucial for maintaining chromatin balance by counteracting histone acetyltransferases, thereby regulating gene transcription [28]. They facilitate the removal of acetyl groups from lysine residues present on histones as well as non-histone proteins, leading to the effective suppression of gene transcription. This process of epigenetic modification serves to uphold genomic stability, thereby securing the meticulous progression of cell development and differentiation [29]. The eighteen mammalian HDACs presently recognized are grouped into four classes according to their similarities in structure, enzymatic functions, and intracellular positions [30].

HDAC inhibitors, recognized as potent epigenetic regulators, have gained significant attention in drug discovery, particularly in the context of cancer [31,32,33]. This stems from the recognition that modulating epigenetic alterations is considered an effective therapeutic approach for cancer, aiming to regulate aberrant transformations. Studies have indicated anti-tumor effects, such as the inhibition of cell growth and differentiation, reduction in angiogenesis, and induction of cell apoptosis, when commonly overexpressed HDACs in cancer are inhibited [34,35,36]. Several HDAC inhibitors have been approved for cancer therapy, primarily demonstrating efficacy in hematological malignancies but yielding less-satisfactory results in the treatment of solid tumors [37,38]. Moreover, HDAC inhibitors have also been revealed to play a role in the anti-inflammatory aspect. This fact has, in turn, expanded clinical research beyond cancer treatment to autoimmune diseases, infectious diseases, and more [39,40]. This can be easily understood, considering that the immune processes occurring in the host in response to cancer may be similar in the context of inflammation [41].

Recently, initial compounds have functioned as non-specific HDAC inhibitors, prompting ongoing research focused on developing inhibitors tailored to the 18 HDAC subtypes. This emphasis stems from recognizing the role of differentiated HDACs in tissue-specific transcriptional control, and the awareness that non-specific HDAC inhibitors may result in undesirable side effects [42]. Our study has established the immunological relevance of HDAC9, which belongs to Class IIa. HDAC9 is frequently overexpressed in cancer cells, rendering it a significant target in cancer therapy research [43,44]. Furthermore, it is associated with the onset of chronic diseases such as cardiovascular diseases, autoimmune diseases, osteoporosis, liver fibrosis, and obesity [45,46]. Therefore, the modulation of HDAC9, either through inhibition or activation, emerges as a promising avenue for therapeutic intervention in various diseases. In the human hematopoietic system, HDAC9 is predominantly expressed in cells of monocytic and lymphoid lineages [42]. HDAC9 plays a role in various immune responses, including the activation of antiviral innate immunity, aligning with our research findings. HDAC9 directly engages with TANK-binding kinase 1 (TBK1), amplifying TBK1 activity to trigger the innate immune antiviral response. Upon exposure to innate immune stimuli, DNA methyltransferase 3A (Dnmt3a) in naïve peritoneal macrophages increases the expression of HDAC9, consequently enhancing host responses [47].

Ripamonti et al. investigated the role of HDAC6 inhibitors in immunity against SARS-CoV-2 [48]. Excessive activation of innate immunity in COVID-19 leads to the overproduction of inflammatory cytokines, resulting in a severe disease course [49]. In this regard, HDAC6 inhibitors were found to reduce cytokine release, decrease T cell exhaustion, and contribute to innate immune cell memory processes, potentially providing therapeutic benefits in severe COVID-19 cases. On the other hand, a study exploring how HDAC inhibition affects SARS-CoV-2 infection in mesothelial cells revealed that blocking HDAC1-3 actually boosts SARS-CoV-2 cell entry, replication, and production [50]. Additionally, another investigation on a different RNA virus, influenza A virus, showed that suppressing HDAC6 activity leads to an elevated virus titer [51]. HDAC6 belongs to HDAC class IIb, while HDAC1-3 is part of class I, and HDAC9, which is in class IIa, encompasses all zinc-dependent members of the largest family of HDACs [52]. Despite their shared characteristics, each class has unique chemical attributes and tissue-specific behaviors. Therefore, insights gained from these previous studies on other classes can aid in understanding the role of HDAC9 identified in this research.

To sum up, the key implication drawn is the necessity for a precise understanding and control of HDAC. While most drugs targeting HDAC have functioned as inhibitors, concentrating on pathological conditions where gene overexpression is a therapeutic target [53,54], our findings reveal that HDAC can contribute positively to the immune response at suitable expression levels, and inhibition might be detrimental [50,51]. This emphasizes the significance of delving into and managing the appropriate expression level and emphasizes the requirement for a precise comprehension of each specific class of HDAC, given its pervasive impact on the entire body.

Our study has several limitations. First, we could not establish causality. Additionally, the findings may not be generalizable to diverse ethnic populations. The selected SNP might not fully represent the entire HDAC9 gene. The lead SNP (rs7795433) is located in the intron region of HDAC9, and there is a notable high recombination rate observed around the SNP cluster. Although there are other genes around the lead SNP, such as TWIST1 and FERD3L, considering the known gene functions, it is presumed that the HDAC9 gene played an important role in immune function related to antibody production. The absence of whole-genome or whole-exome sequencing limits the discovery of other influential genes. Moreover, this study is constrained by not selecting high- and low-antibody groups from a larger population. Lastly, individuals who had contracted COVID-19 were excluded during the study to mitigate the impact of potential natural infections on antibody concentration after vaccination. Consequently, the actual efficacy difference in COVID-19 prevention could not be assessed. Nevertheless, our study is significant in identifying candidate gene HDAC9, which is associated with differences in antibody production after COVID-19 mRNA vaccination, focusing on the Korean population. Furthermore, by observing higher HDAC9 gene expression in the high-antibody group within the same gene, we have confirmed that the degree of HDAC9 expression can influence differences in antibody generation after vaccination. This information is considered crucial for future vaccine administration planning and personalized immune management.

## 5. Conclusions

The HDAC9 gene has been identified to play a role in antibody generation after COVID-19 vaccination, with a higher likelihood of increased antibody production in the AA allele group. These findings have the potential not only to inform individual vaccination schedules for future cases of COVID-19 but also to contribute to the development of therapeutic interventions as epigenetic modulators.

## Figures and Tables

**Figure 1 vaccines-12-00626-f001:**
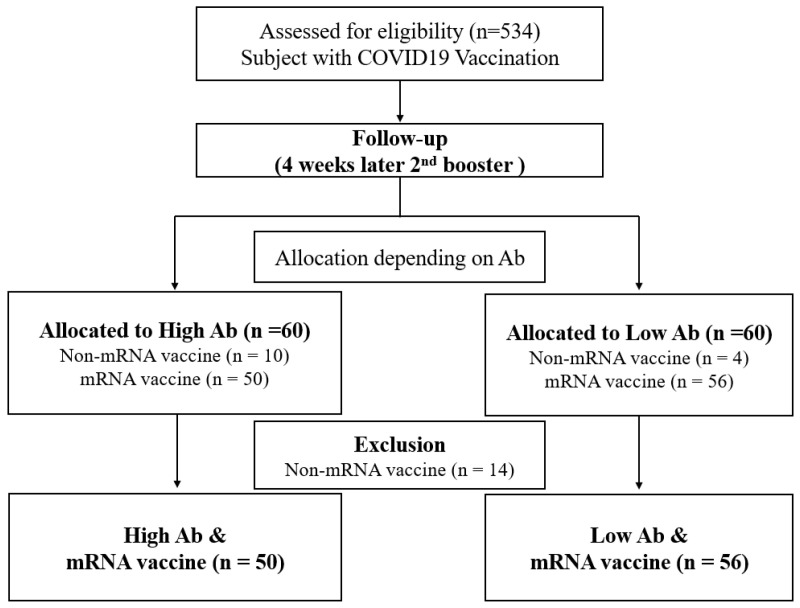
Flowchart for the selection of study participants.

**Figure 2 vaccines-12-00626-f002:**
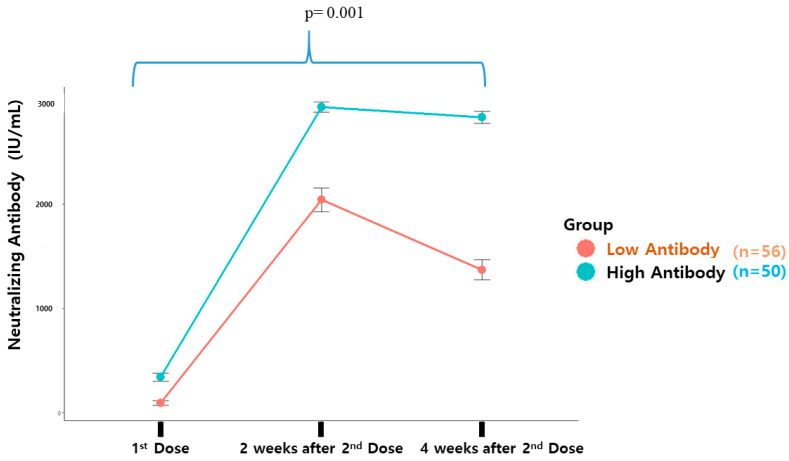
Linear mixed-effects model indicating the fixed effect for vaccination, group, and group-by-vaccination interaction.

**Table 1 vaccines-12-00626-t001:** Baseline characteristics of the study population according to antibody level.

		High Ab	Low Ab	*p*-Value
N		50	56	
Age, years		45.9 ± 8.1	46.9 ± 7.5	0.489
Sex (male, %)		24 (48.0)	33 (58.9)	0.035
Body mass index, kg/m^2^		26.4 ± 4.3	23.9 ± 3.1	0.001
Glucose, mg/dL		95.6 ± 19.1	99.5 ± 15.9	0.256
Total cholesterol, mg/dL		182.4 ± 48.7	187.0 ± 40.0	0.597
Triglyceride, mg/dL		118.5 (80.2–203.7)	127.5 (66.7–196.5)	0.482
HDL cholesterol, mg/dL		47.6 ± 15.6	54.2 ± 14.4	0.024
LDL cholesterol, mg/dL		119.1 ± 42.0	119.1 ± 38.6	0.990
Ig E		69.4 (26.1–170.7)	47.0 (18.9–102.0)	0.055
Vitamin D		21.4 (16.5–26.9)	19.8 (13.9–23.3)	0.430
WBC ( $\times 10^{3}$ L)		6.24 ± 1.62	5.65 ± 1.31	0.040
AST (IU/L)		28.9 ± 10.2	22.5 ± 9.7	0.166
ALT (IU/L)		26.7 ± 18.0	24.2 ± 16.8	0.463
Comorbid condition, n (%)				
Hypertension, (%)		6 (12.0)	5 (8.9)	0.604
Diabetes, (%)		4 (8)	4 (7.1)	0.867
Neutralizing Ab Titer				
First vaccination		342.3 ± 269.0	92.7 ± 168.7	0.001
Second vaccination		2935.2 ± 334.3	2044.0 ± 857.9	0.001
4 Weeks later		2837.9 ± 410.3	1373.3 ± 707.1	0.001
Vaccination, n (%)				
First dose	Second dose			0.001
Moderna	Moderna	29 (58.0)	12 (21.4)	
Pfizer	Pfizer	21 (42.0)	44 (78.6)	

Data are expressed as the mean ± SD or percentage. *p*-values were calculated using the independent two sample *t*-test or the chi-squared test.

**Table 2 vaccines-12-00626-t002:** Top 10 SNPs identified in the genome-wide association study for low- and high-antibody production following COVID-19 vaccination.

						Model 1		Model 2	
Chr ^a^	Gene	BP ^b^	SNP ^c^	Minor Allele	MAF	OR (L95–U95) ^d^	*p*-Value	OR (L95–U95) ^d^	*p*-Value
7	*HDAC9*	18833779	rs7795433	G	0.48	5.63 (2.51–12.65)	3 × 10^−5^	6.99 (2.74–17.87)	5 × 10^−5^
8	*MSC-AS1*	71979513	rs10111413	A	0.50	0.24 (0.12–0.47)	5 × 10^−5^	0.21 (0.09–0.47)	1 × 10^−4^
7	*HDAC9*	18828330	rs6951522	C	0.47	5.18 (2.37–11.36)	4 × 10^−5^	6.76 (2.69–16.96)	5 × 10^−5^
7	*HDAC9*	18838251	rs2073963	G	0.48	5.14 (2.35–11.24)	4 × 10^−5^	6.24 (2.53–15.37)	7 × 10^−5^
4	*NDST4/MIR1973*	115308340	rs118002192	C	0.13	0.12 (0.04–0.38)	4 × 10^−4^	0.17 (0.05–0.60)	6 × 10^−3^
11	*HTR3A*	113984620	rs1176717	A	0.21	0.23 (0.10–0.54)	7 × 10^−4^	0.31 (0.12–0.79)	1 × 10^−2^
7	*AUTS2*	70660723	rs17141963	T	0.47	0.20 (0.17–0.63)	8 × 10^−4^	0.26 (0.12–0.59)	1 × 10^−3^
7	*HDAC9*	18837993	rs957958	G	0.50	4.42 (2.12–9.22)	8 × 10^−5^	5.04 (2.20–11.56)	1 × 10^−4^
6	*LOC105377862*	3491893	rs7742726	C	0.36	3.73 (1.69–8.25)	1 × 10^−3^	3.58 (1.52–8.44)	4 × 10^−3^
7	*HDAC9*	18837785	rs957960	A	0.47	5.02 (2.29–11.00)	6 × 10^−5^	6.23 (2.51–15.45)	8 × 10^−5^

^a^ Chromosome. ^b^ Base pair. ^c^ Top SNPs with *p*-values less than 1 × 10^−3^ in the analysis after adjustment for age and sex. ^d^ Odds ratios (ORs) and confidence interval. Usually, the lead SNP in each locus is reported after pruning with MAF. Covariants: Model 1: age, sex, BMI; Model 2: age, sex, BMI, vaccine type, PC1, PC2.

**Table 3 vaccines-12-00626-t003:** Individual or allelic proportion of HDAC9 SNP (rs7795433) in the high- and low-antibody production groups, respectively.

Antibody Production	rs7795433 Genotype				rs7795433 Allele		
	AA	GA	GG	Total (n)	A	G	Total (n)
High (%)	Group A	Group B	Group C		Effect Alle	Non-Effect Alle	
	16 (32%)	26 (52%)	8 (16%)	50	58 (58%)	42 (42%)	100
Low (%)	Group D	Group E	Group F				
	12 (21%)	30 (54%)	14 (25%)	56	54 (48%)	58 (52%)	112

**Table 4 vaccines-12-00626-t004:** Analysis of the relative expression of HDAC9 genes between high- and low-antibody production groups within the same genotype group.

Average Delta ct per Sample (Hdac9–GapDH)						
	AA		GA		GG	
	MM		Mm		mm	
	Group A		Group B		Group C	
R374	4.088497	R082	5.982103	R035	NA	
R434	4.125577	R362	4.662499	R391	NA	
R503	3.808289	R436	5.774843	R498	4.644077	
	Group D		Group E		Group F	
R501	5.253894	R038	6.569895	R360	4.760575	
R383	5.220209	R386	6.568871	R365	4.2658	
R471	4.61415	R415	6.731386	R504	4.924133	
**Average Delta ct for Each Group (Hdac9–GapDH)**						
	MM		Mm		mm	
	Group A		Group B		Group C	
Mean and SD	4.007454	0.173476	5.473148	0.70965	4.644077	0
	Group D		Group E		Group F	
Mean and SD	5.029417	0.360026	6.623384	0.093534	4.650169	0.342772
**Relative Quantitative Comparison between Each Group**						
	A vs. D		B vs. E		C vs. F	
	MM		Mn		mm	
Delta delta CT	−1.02196		−1.15024		−0.00609	
RQ Ct (relative quantification)	2.03068		2.219501		1.004232	
*t*-test *p*-value	0.011429		0.04965		0.09891	

## Data Availability

The data presented in this study are available on request from the corresponding author.

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
