# Peer review of "The Human Genetic Differences in the Outcomes of mRNA Vaccination against COVID-19: A Prospective Cohort Study"

_vaccines, 2024, doi:10.3390/vaccines12060626_

Round 1

Reviewer 1 Report

Comments and Suggestions for Authors

After immunization with mRNA vaccine, the AA genotype population with the HDAC9 gene seems to have a high level of anti COVID-19 antibody by GWAS in this manuscript. 

Special comment:

1. The other SNPs of the HDAC9 gene showed good association with COVID-19 vaccination with OR>4 sides the rs7795433 SNPs of the HDAC9 gene

2. The SNP of the LOC105377862 gene also showed good association with COVID-19 vaccination with OR>3.73 Why not analyze?

3. Is the objectivity rate higher among participants in the low antibody group? This affects the high antibody levels in the population with high BMI as shown in Table 1.

4. Where is Table 4 in section 3.4?

Reviewer 2 Report

Comments and Suggestions for Authors

Ryu et al. investigated how individual genetic variations influence the neutralizing activity following mRNA vaccination and underscored the essential role of vaccination in curbing the spread of COVID-19 while highlighting the necessity to ensure vaccine efficacy amid genetic diversity. The study presents compelling findings that individuals with the AA genotype (HDAC9 gene’s rs7795433 SNP) are likely to produce higher antibody levels post-vaccination, whereas those with the GG genotype may exhibit the opposite outcome. These insights are crucial for understanding the genetic underpinnings that may affect vaccine responsiveness.

This is an interesting paper, I have a few suggestions.

1. Including units on the Y-axis of Figure 2 would be beneficial for better clarity and to aid the reader's understanding.

2. The discussion touches upon related studies focusing on HDAC1-3 and HDAC6. I recommend a more detailed exploration of the relationship between HDAC9 and HDAC1-3, as well as HDAC6, especially since HDAC9 is identified as one of the top 10 highly relevant SNPs in this study.

3. There is a concern regarding the potential for substantial error in the antibody detection methods. Considering alternative detection schemes could enhance the reliability of the results.

Comments on the Quality of English Language

Minor editing of English language required.

Author Response

  1. Including units on the Y-axis of Figure 2 would be beneficial for better clarity and to aid the reader's understanding.

Answer) We have revised the figure according to your advice.

  1. The discussion touches upon related studies focusing on HDAC1-3 and HDAC6. I recommend a more detailed exploration of the relationship between HDAC9 and HDAC1-3, as well as HDAC6, especially since HDAC9 is identified as one of the top 10 highly relevant SNPs in this study.

Answer) We have supplemented the discussion content as per your suggestion.

“HDAC6 belongs to HDAC class IIb, while HDAC1-3 are part of class I, and HDAC9, which is in class IIa, are all zinc-dependent members of the largest family of HDACs [51]. Despite their shared characteristics, each class has unique chemical attributes and tissue-specific behaviors. Therefore, insights gained from these previous studies on other classes can aid in understanding the role of HDAC9 identified in this research.”

  1. There is a concern regarding the potential for substantial error in the antibody detection methods. Considering alternative detection schemes could enhance the reliability of the results.

Answer) The reliability of our test results has been confirmed using two levels of quality control substances for each run. Additionally, we reviewed the patient's history and correlated anti-S, anti-N, and neutralization antibodies, demonstrating a very good correlation. The reagent used in our test is FDA-approved and does not require an additional test, except for unavoidable random errors.

Round 2

Reviewer 2 Report

Comments and Suggestions for Authors

Accept in present form.

Comments on the Quality of English Language

 Minor editing of English language required.